# Silica-Coated Magnetic Nanoparticles Decrease Human Bone Marrow-Derived Mesenchymal Stem Cell Migratory Activity by Reducing Membrane Fluidity and Impairing Focal Adhesion

**DOI:** 10.3390/nano9101475

**Published:** 2019-10-17

**Authors:** Tae Hwan Shin, Da Yeon Lee, Abdurazak Aman Ketebo, Seungah Lee, Balachandran Manavalan, Shaherin Basith, Chanyoung Ahn, Seong Ho Kang, Sungsu Park, Gwang Lee

**Affiliations:** 1Department of Physiology, Ajou University School of Medicine, Suwon 16499, Korea; catholicon@ajou.ac.kr (T.H.S.); ekdus93@ajou.ac.kr (D.Y.L.); bala.msuman@gmail.com (B.M.); shaherinb@gmail.com (S.B.); 2School of Mechanical Engineering, Sungkyunkwan University, Suwon 16419, Korea; anuabman@gmail.com (A.A.K.); nanopark@skku.edu (S.P.); 3Department of Applied Chemistry and Institute of Natural Sciences, Kyung Hee University, Yongin-si 17104, Korea; moon11311@naver.com (S.L.); shkang@khu.ac.kr (S.H.K.); 4Department of Molecular and Cell Biology, University of California, Berkeley, CA 94720, USA; justinahn99@berkeley.edu

**Keywords:** magnetic nanoparticles, human bone marrow-derived mesenchymal stem cells, membrane fluidity, focal adhesion, cytoskeletal abnormality

## Abstract

For stem cell-based therapies, the fate and distribution of stem cells should be traced using non-invasive or histological methods and a nanomaterial-based labelling agent. However, evaluation of the biophysical effects and related biological functions of nanomaterials in stem cells remains challenging. Here, we aimed to investigate the biophysical effects of nanomaterials on stem cells, including those on membrane fluidity, using total internal reflection fluorescence microscopy, and traction force, using micropillars of human bone marrow-derived mesenchymal stem cells (hBM-MSCs) labelled with silica-coated magnetic nanoparticles incorporating rhodamine B isothiocyanate (MNPs@SiO_2_(RITC)). Furthermore, to evaluate the biological functions related to these biophysical changes, we assessed the cell viability, reactive oxygen species (ROS) generation, intracellular cytoskeleton, and the migratory activity of MNPs@SiO_2_(RITC)-treated hBM-MSCs. Compared to that in the control, cell viability decreased by 10% and intracellular ROS increased by 2-fold due to the induction of 20% higher peroxidized lipid in hBM-MSCs treated with 1.0 µg/µL MNPs@SiO_2_(RITC). Membrane fluidity was reduced by MNPs@SiO_2_(RITC)-induced lipid oxidation in a concentration-dependent manner. In addition, cell shrinkage with abnormal formation of focal adhesions and ~30% decreased total traction force were observed in cells treated with 1.0 µg/µL MNPs@SiO_2_(RITC) without specific interaction between MNPs@SiO_2_(RITC) and cytoskeletal proteins. Furthermore, the migratory activity of hBM-MSCs, which was highly related to membrane fluidity and cytoskeletal abnormality, decreased significantly after MNPs@SiO_2_(RITC) treatment. These observations indicated that the migratory activity of hBM-MSCs was impaired by MNPs@SiO_2_(RITC) treatment due to changes in stem-cell biophysical properties and related biological functions, highlighting the important mechanisms via which nanoparticles impair migration of hBM-MSCs. Our findings indicate that nanoparticles used for stem cell trafficking or clinical applications should be labelled using optimal nanoparticle concentrations to preserve hBM-MSC migratory activity and ensure successful outcomes following stem cell localisation.

## 1. Introduction

Nanoparticles are being increasingly used for disease diagnosis and therapy and cell tracing [1,2,3]. Among nanoparticles, magnetic nanoparticles (MNPs) and MNPs coated with biocompatible polymers and silica for safety are used for in vitro cell labelling, fluorescence-based in vivo cell tracking, and magnetic resonance imaging (MRI)-based stem cell-labelled in vivo tracing [4,5,6,7]. However, detailed information regarding the biophysical effects of nanoparticles at the cellular level is still limited.

Mesenchymal stem cells (MSCs) are used in biomedical applications (cytotherapy) for multiple sclerosis and for cardiovascular, ischemic, and neurodegenerative disorders [8,9,10,11,12]. In particular, bone marrow-derived MSCs (BM-MSCs) possess useful characteristics, including high degree of plasticity, trophic factor secretion, and immune response suppression capability [13,14,15]. Thus, human BM-MSCs (hBM-MSCs) are considered promising therapeutic candidates for clinical application [12,16,17,18]. For successful cytotherapeutic outcomes using stem cells [19,20], nanoparticle-based methods for tracking the localization of transplanted cells in the body are essential to ensure their distribution in the impaired tissue [21]. However, nanoparticle-induced biophysical disturbances caused by reactive oxygen species (ROS) generation, which result in changes in normal physiological redox-regulated functions and cellular alteration, are matters of concern [22]. For example, silica-coated magnetic nanoparticles incorporating rhodamine B isothiocyanate (MNPs@SiO_2_(RITC)) induce ROS production, leading to endoplasmic reticulum (ER) stress, reduced proteasome activity, and altered cellular metabolism [23,24,25].

Nanoparticle-induced ROS oxidize proteins to generate protein radicals [22] and induce lipid peroxidation [26], which impairs the functions of the plasma membrane [27,28]. These oxidative cleavage events deplete unsaturated phospholipids and cholesterol in the cell membrane, leading to loss of the fluidity and permeability of the membrane and, thereby, affecting its physiological functions [29,30,31]. Several studies have reported that lipid peroxidation decreases membrane fluidity, indicating that the membrane fluidity of nanoparticle treated-cells can change due to oxidative stress-induced membrane damage [30,32]. However, the relationship between nanoparticle-induced lipid peroxidation and membrane fluidity remains unclear.

The membrane and the cytoskeleton are tightly linked via phosphoinositides, especially via focal adhesion and actin assembly [33,34,35]. Therefore, membrane damage, caused by nanoparticle-induced oxidative stress, is highly related to the cytoskeleton. Cell morphology is determined by the balance between adhesion and tension. Nanoparticles disrupt the cytoskeleton to affect focal adhesion proteins and, thus, adhesion [36] initiated by the lamellipodia (branched actin filaments) and filopodia (extended finger-like protrusions) as focal complexes [37]. During cell death; membrane repair; and osmotic-, oxidative-, and heat stress; these structures are abolished, such that cells display a rounded and shrunken morphology [38,39]. However, studies on the effect of nanoparticles on cell adhesion and tension are limited.

The focal adhesion of hBM-MSCs is strongly associated with changes in cellular traction forces [40]. Elastomeric pillar arrays are considered excellent for measuring cellular traction force by calculating the nanometric level of pillar deflection [41,42]. In addition, sub-micron pillar arrays have been shown to mimic continuous substrates of specific rigidity [43]. Thus, biophysical changes in nanoparticle-treated cells have been quantitatively studied using elastomeric submicron pillars [41,43].

In this study, we aimed to investigate the biophysical properties of MNPs@SiO_2_(RITC)-treated hBM-MSCs, such as membrane fluidity (using total internal reflection fluorescence microscopy (TIRFM)), traction force (using micropillars), cytoskeletal characteristics, and migratory activity.

## 2. Materials and Methods

### 2.1. MNPs@SiO_2_(RITC) and Silica Nanoparticles (NPs)

MNPs@SiO_2_(RITC) particles, composed of a ~9 nm cobalt ferrite core (CoFe_2_O_3_) chemically bonded to rhodamine isothiocyanate dye (RITC) and coated by a silica shell [4], were purchased from BITERIALS (Seoul, Korea). Previously, these nanoparticles have been characterized for confirming their quality [44]. Size distribution and morphology are important factors determining the uniformity of nanoparticles and were analyzed using electron and atomic microscopy [44]. Hydrodynamic size, polydispersity, and surface charge were determined using dynamic light scattering [45]. The purity and contents of nanoparticles are usually analyzed using an X-ray based technique [44]. In this study, X-ray diffraction (XRD) analysis using a high-power X-Ray diffractometer (Ultima III, Rigaku, Japan) confirmed the structure of MNPs@SiO_2_(RITC) (data not shown). The silica NPs were composed of identical materials and were of a similar size as the MNPs@SiO_2_(RITC) shell, and their biological effects were similar to those of MNPs@SiO_2_(RITC) [23,24,46,47]. The diameters of the MNPs@SiO_2_(RITC) and silica NPs were 50 nm, and the *zeta* potential of MNPs@SiO_2_(RITC) was between −40 to −30 mV [4,46]. A previous study determined ~10^5^ particles of MNPs@SiO_2_(RITC) per cell in MNPs@SiO_2_(RITC)-treated MCF-7 cells using inductively coupled plasma atomic emission spectrometry [4]. Furthermore, in previous reports, the dosage was determined by measuring the fluorescence intensity of HEK293 cells treated with MNPs@SiO_2_(RITC) at concentrations ranging from 0.01 to 2.0 μg/μL for 12 h. The optimal concentration of MNPs@SiO_2_(RITC) was 0.1 µg/µL for in vitro use, whereas 1.0 µg/µL was the plateau concentration for cellular uptake [24]. Furthermore, MNPs@SiO_2_(RITC) concentrations ranging from 0 to 1.0 μg/μL have been used for MRI contrasting without toxicological effects on human cord blood-derived MSCs [48], and caused changes in gene expression and metabolic profiles similar to those of the control HEK293 cells at 0.1 µg/µL [24]. In addition, the uptake efficiency of MNPs@SiO_2_(RITC) almost plateaued at 1.0 µg/µL in HEK293 cells [24,25]. The dose-dependent fluorescence intensity of MNPs@SiO_2_(RITC)-labelled hBM-MSCs was similar to those of labelled HEK293 cells. In addition, the viability of human cord blood-derived MSCs was determined to assess the cytotoxic effect of MNPs@SiO_2_(RITC) after 24, 48, and 72 h of treatment with 0–1.0 µg/µL MNPs@SiO_2_(RITC); compared to the control group, no significant cytotoxic effect was observed [48]. Therefore, in this study, hBM-MSCs were treated with 0.1 µg/µL (low dose) MNPs@SiO_2_(RITC)or 1.0 µg/µL (high dose), similarly to previous reports [23,24,47].

### 2.2. Cell Culture

hBM-MSCs were purchased from PromoCell (Heidelberg, Germany) and were cultured as described in previous studies [49,50]. Briefly, the cells were rinsed with phosphate buffered saline (PBS), resuspended, cultured in Dulbecco’s low-glucose modified Eagle’s medium (DMEM, Gibco, Grand Island, NY, USA) supplemented with 10% fetal bovine serum (Gibco, Grand Island, NY, USA), 100 units/mL penicillin, and 100 μg/mL streptomycin (Gibco, USA), and incubated in a 5% humidified CO_2_ chamber at 37 °C. The hBM-MSC surface markers, CD73 and CD105, and negative markers of hBM-MSCs, namely, CD34 and CD45, were analyzed and maintained (data not shown).

### 2.3. Morphological Analysis of hBM-MSCs

To evaluate the MNPs@SiO_2_(RITC)-induced morphological changes, hBM-MSCs were treated with 0.1 and 1.0 µg/µL of MNPs@SiO_2_(RITC) for 12 h. Images were acquired with an Axio Vert 200M fluorescence microscope (Zeiss, Jena, Germany). The excitation wavelength for MNPs@SiO_2_(RITC) was 530 nm.

### 2.4. Cell Viability Assay

For analysis of cell viability, the CellTiter 96-cell proliferation assay kit (MTS, Promega, Madison, WI, USA) was used, according to the manufacturer’s instructions. Briefly, 2 × 10^4^ hBM-MSCs were seeded on 96-well assay plates. After 16 h, the hBM-MSCs were washed with PBS and treated with MNPs@SiO_2_(RITC) for 12 h. The hBM-MSCs were then washed with PBS to remove excess MNPs@SiO_2_(RITC), and MTS solution was added to each well (1/10 volume of media). Subsequently, the plate was incubated for 1 h in a 5% CO_2_ chamber maintained at 37 °C. The absorbance of the soluble formazan was measured using a plate reader (Molecular Devices, San Jose, CA, USA) at 490 nm. Values were normalized relative to the protein absorbance value for each corresponding group.

### 2.5. Evaluation of Intracellular ROS Levels in hBM-MSCs

Intracellular ROS levels in hBM-MSCs were evaluated using DCFH-DA staining (Cell Biolabs, San Diego, CA, USA) according to the manufacturer’s protocol. Briefly, control and MNPs@SiO_2_(RITC)-treated hBM-MSCs were resuspended in 10 µM DCFH-DA and incubated in a 5% CO_2_, 37 °C chamber for 1 h. The hBM-MSCs were washed twice with PBS, and DCF fluorescence was measured using a fluorescence microplate reader (Gemini EM, Molecular Devices, Sunnyvale, CA, USA) at 480 nm excitation and 530 nm emission wavelengths.

### 2.6. Evaluation of Lipid Peroxidation

Peroxidized lipids in control and MNPs@SiO_2_(RITC)-treated hBM-MSCs were quantified using a lipid peroxidation kit (Cayman Chemical, Ann Arbor, MI, USA) according to the manufacturer’s instructions. Briefly, hBM-MSCs were treated with MNPs@SiO_2_(RITC) for 12 h. Subsequently, hBM-MSCs were trypsinized and washed twice with PBS. The hBM-MSCs were transferred to glass tubes, and the lipids were extracted in crystalline solid-saturated methanol and cold chloroform. The mixture was centrifuged (1500× *g*, 0 °C, 5 times) to form two layers, and the bottom chloroform layer was collected. The samples were mixed with 2.25 mM ferrous sulphate, 0.1 M hydrochloric acid, and 1.5% ammonium thiocyanate in methanol at a 9:1 ratio. Next, the mixtures were incubated at room temperature for 5 min. Ferric ions were produced in the reaction, and the level of peroxidized unsaturated lipids was evaluated using thiocyanate as a chromogen. Absorbance was measured at 500 nm using a quartz cuvette and microplate reader (Molecular Devices, San Jose, CA, USA).

### 2.7. Measurement of Membrane Fluidity

Membrane fluidity of hBM-MSCs was measured using a homemade combined differential interference contrast (DIC)-total internal reflection fluorescence microscopy (TIRFM) experimental system [49,51]. The procedure was based on a previously described protocol [52,53]. Briefly, hBM-MSCs were cultured on 0.13–0.16 mm thick cover slips and treated with MNPs@SiO_2_(RITC) for 12 h. The hBM-MSCs were incubated with media containing 10 µM Laurdan in a 5% CO_2_, 37 °C chamber for 2 h. The hBM-MSCs were washed twice with PBS and incubated with fixation buffer (Cytofix; BD, San Jose, CA, USA). Subsequently, the hBM-MSC-containing cover slips were mounted onto 0.13–0.16 mm thick cover slips with mounting medium (Prolong Gold antifade; Molecular Probes, Eugene, OR, USA). Laurdan fluorescence was observed with a 100× objective lens (oil-type, Olympus UPLFL 1.3 N.A., W.D. 0.1 mm) and an A CCD camera (QuantEM 512SC, Photometrics, Tucson, AZ, USA). The fluorescence intensity was measured using an excitation wavelength of 405 nm, and emission fluorescence was detected using 420 nm and 473 nm bandpass filter (resolution: ±5 nm). As a parameter of membrane fluidity, the generalized polarizations (GP) = (fluorescence intensity at 420 nm—fluorescence intensity at 473 nm)/(fluorescence intensity at 420 nm + fluorescence intensity at 473 nm) was calculated, and pseudo-colored GP value images were generated using the Image J 1.48v software (NIH, Bethesda, MD, USA) [54]. Gauss distributions were generated using the nonlinear fitting algorithm in Sigma Plot 10.0 (Systat Software Inc., San Jose, CA, USA).

### 2.8. Immunocytochemistry

hBM-MSCs were seeded on cover slips and treated with 0.1 μg/μL and 1.0 μg/μL MNPs@SiO_2_(RITC) for 12 h. The cells were, then, fixed in Cytofix buffer (BD, San Jose, CA, USA). To reduce non-specific binding, the cover slips were blocked with PBS containing 2% bovine serum albumin (BSA) and 0.1% Triton-X100 (Sigma-Aldrich, St Louis, MO, USA). For actin labelling, hBM-MSCs were then incubated with Alexa Fluor 488-conjugated phalloidin (Molecular Probe, Carlsbad, CA, USA, 1:200), diluted in blocking buffer, for 1 h at room temperature. For β-tubulin and vinculin labelling, hBM-MSCs were incubated with anti-β-tubulin mouse polyclonal antibody (Molecular Probes, Carlsbad, CA, USA, 1:200) or anti-vinculin rabbit monoclonal antibody (Becton Dickinson, Franklin Lakes, NJ, USA, 1:200) diluted in blocking buffer, for 12 h at 4 °C. Following three times rinsing in PBS containing 0.1% Triton-X100, the cells were incubated with Alexa Fluor 488-conjugated anti-mouse goat polyclonal antibody or Alexa Fluor 647-conjugated anti-rabbit goat polyclonal antibody for 1 h at room temperature. The labelled cells were washed thrice with PBS containing 0.1% Triton-X100 and incubated with PBS containing 10 µg/mL Hoechst 33342 for 10 min at room temperature to label the nuclei. After washing three times with PBS, cover slips were mounted onto slides using Prolong Gold antifade mounting medium (Molecular Probes). Fluorescent images were acquired using confocal laser scanning microscopy (LSM710, Carl Zeiss Microscopy GmbH, Jena, Germany). The excitation wavelengths for Alexa Fluor 488, Hoechst 33342, and MNPs@SiO_2_(RITC) were 488, 405, and 530 nm, respectively. The attached hBM-MSCs areas were analyzed using the Image J software.

### 2.9. Western Blotting

hBM-MSCs were seeded at a density of 2 × 10^5^ cells/well in 6-well plates and cultured for 36 h. hBM-MSCs were treated with 0.1 or 1.0 µg/µL MNPs@SiO_2_(RITC) for 12 h and lysed in radioimmunoprecipitation assay (RIPA) buffer. The lysates were vortexed and incubated at 4 °C for 1 h. Then, the lysates were centrifuged at 14,000× *g* for 15 min at 4 °C, and the supernatants were collected. Next, 40 µg protein was subjected to sodium dodecyl sulphate-polyacrylamide gel electrophoresis (SDS–PAGE) and transferred onto nitrocellulose membranes. The membranes were blocked with 3% non-fat milk for 1 h at room temperature and incubated with primary antibody overnight at 4 °C. The following primary antibodies were used: t-c-SRC (1:2000, Santa Cruz Technologies, Santa Cruz, CA, USA), p-c-SRC (1:2000, Santa Cruz Technologies, Santa Cruz, CA, USA), t-FAK (1:2000, Santa Cruz Technologies, Santa Cruz, CA, USA), p-FAK (1:2000, Santa Cruz Technologies, Santa Cruz, CA, USA), and β-actin (1:5000, Cell Signaling, La Jolla, CA, USA). Secondary antibodies were used at a dilution of 1:2000 (Santa Cruz Technologies, USA). The blots were developed using enhanced chemiluminescence solution (ECL, Thermo Scientific, Waltham, MA, USA), and luminescence was captured on medical blue X-ray film (AGFA, Mortsel, Belgium) in a dark room.

### 2.10. Microfabrication of Pillar Arrays

A standard photolithograph was used to fabricate a mold with arrays of holes over a silicon wafer [55]. To fabricate pillar arrays, Polydimethylsiloxane (PDMS) was mixed at 10:1 ratio with its curing agent (Sylgard 184; Dow Corning, Midland, MI, USA) and degassed for 15 min. Next, it was spin-coated over the mold, degassed again for 30 min to remove trapped air bubbles within the mixture, and cured at 80 °C for 4 h and 30 min until the Young modulus of the PDMS reached 2 ± 0.1 MPa. Subsequently, the cured pillar array was carefully removed from the mold. In the case of micron scale pillars, cellular contractions occurred around individual pillars (diameter 2 µm) [43,56]. Thus, we generated linearly arranged pillar arrays of 900 nm diameter, 1 µm height, and a pillar diameter two times the center-to-center distance between pillars. The bending stiffness of the pillar was calculated based on Euler-Bernoulli beam theory [57]:
$$k=\frac{3}{64}\pi E\frac{D^{4}}{L^{3}}$$
where *D* is the diameter, *L* is the length, and *E* is the Young modulus of the pillar [43]. The bending stiffness (*k*) of the pillar arrays used in this study was 28.8 nN/µm.

### 2.11. Measurement of Traction Force

Images of pillars in hBM-MSCs were captured using a live cell chamber at 1 Hz in a fluorescence microscope (Deltavision, GE Healthcare, Chicago, IL, USA) equipped with a camera (CoolSNAP HQ^2^, Photometrics) at 37 °C and 5% humidity. The place of each pillar in each frame was determined using the pillar tracking plugin (PillarTracker 1.1.3 version) of the Image J software. PillarTracker uses the pillar reconstruction algorithm to establish an exact grid of the pillar arrays, thus allowing users to automatically detect and track the locations of the pillars. Throughout this study, pillars with no cell contact were used as reference pillars. To account for stage drift, the average displacement of the reference pillars was deducted from the displacement data of pillars deflected by hBM-MSCs. To avoid unwanted displacement of pillars by MNPs@SiO_2_(RITC), MNPs@SiO_2_(RITC)-treated cells were washed five times using Dulbecco’s phosphate buffered saline (DPBS) before seeding on a pillar array. The displacement of each pillar was multiplied by its bending stiffness to calculate the traction force.

### 2.12. Wound Healing Assay

hBM-MSCs were seeded and grown to 100% confluence in 6-well plates, followed by washing in PBS. Cell monolayers were wounded with a 200-µL micropipette tip in two different places in each well, treated with MNPs@SiO_2_(RITC) in serum-free media, and allowed to migrate for 16 h. Images of the wounded areas were captured under an Axio Vert 200M fluorescence microscope (Zeiss, Jena, Germany) at 0 and 16 h. The excitation wavelength for MNPs@SiO_2_(RITC) was 530 nm. Migration activity was quantified by analyzing the cell number.

### 2.13. Invasion Assay

Invasion assays of hBM-MSCs were performed using an 8-µm pore size transwell polycarbonate membrane (Corning, Corning, NY, USA). The upper side of the membrane was coated with Matrigel (1:10 dilution in 0.01 M Tris pH 8.0, 0.7% NaCl) for 2 h at 37 °C. hBM-MSCs were treated with MNPs@SiO_2_(RITC) for 12 h. Next, 2.5 × 10^4^ hBM-MSCs were transferred to the upper chamber of the transwell in serum-free media, and 10% FBS containing medium was added to the lower chamber as a chemoattractant. The cells were incubated for 12 h at 37 °C. Subsequently, the cells on the upper side of the membrane were removed with a cotton swab, and the invading cells on the lower side of the membrane were fixed in Cytofix buffer (BD, San Jose, CA, USA) and stained with Hoechst 33342. Images were acquired using an Axio Vert 200M fluorescence microscope (Zeiss, Jena, Germany). The excitation wavelengths for MNPs@SiO_2_(RITC) and Hoechst 33342 were 530 nm and 405 nm, respectively. The number of invading cells was counted using the Image J software.

### 2.14. Statistical Analysis and Error Correction

The results were analyzed using one-way analysis of variance (ANOVA) with Bonferroni’s multiple-comparison test of the IBM-SPSS software (IBM Corp., Armonk, NY, USA). Differences were considered significant for *p* values < 0.05. In the micropillar experiments, errors of the pillar deflections were corrected by reducing the average deflection of pillars outside the cell.

## 3. Results

### 3.1. Decrease in Cell Viability and ROS Generation of MNPs@SiO_2_(RITC)-Treated hBM-MSCs

To evaluate the viability of and ROS generation in MNPs@SiO_2_(RITC)-treated hBM-MSCs, hBM-MSCs were treated with MNPs@SiO_2_(RITC) for 12 h before analysis (Figure 1a). A monolayer of hBM-MSCs was clearly observed for the non-treated control cells, while the monolayer was disintegrated for the MNPs@SiO_2_(RITC)-treated hBM-MSCs (Figure 1b). Furthermore, compared to that in the non-treated control, the viability of hBM-MSCs decreased by ~10% upon treatment with 0.1 and 1.0 µg/µL MNPs@SiO_2_(RITC) (Figure 1c). However, the viability of cells treated with 0.1 µg/µL MNPs@SiO_2_(RITC) was not statistically significantly different from that of cells treated with 1.0 µg/µL MNPs@SiO_2_(RITC). Intracellular ROS generation was evaluated using 2′,7′-dichlorodihydrofluorescin diacetate (DCFH-DA) staining in MNPs@SiO_2_(RITC)-treated hBM-MSCs. Treatment with MNPs@SiO_2_(RITC) and 50 nm-sized silica NPs increased intracellular ROS levels (Figure 1d,e). In particular, compared to that in non-treated control cells and cells treated with 0.1 µg/µL NPs, intracellular ROS levels increased by more than 50% in cells treated with 1.0 µg/µL MNPs@SiO_2_(RITC) and silica-NPs. Furthermore, the level of ROS-induced peroxidized lipids increased by ~30% in hBM-MSCs treated with 1.0 µg/µL MNPs@SiO_2_(RITC) (Figure 1f).

### 3.2. Reduction in Membrane Fluidity of hBM-MSCs after MNPs@SiO_2_(RITC) Treatment

To analyze changes in membrane fluidity due to peroxidation of lipids, the MNPs@SiO_2_(RITC)-treated hBM-MSCs were stained with Laurdan and generalized polarization (GP) values were calculated using TIRFM (Figure 2a). The number of high-GP areas on the hBM-MSC surface—Corresponding to rigid domains—Increased upon MNPs@SiO_2_(RITC) treatment. In particular, the abundantly distributed region of MNPs@SiO_2_(RITC) majorly co-localized with the high GP-distribution region in a GP scale of −1.0 to 1.0 (Figure 2b). GP frequency distribution values of MNPs@SiO_2_(RITC) treated-hBM-MSCs were subtracted from the corresponding values of the non-treated control hBM-MSCs to obtain frequency difference curves (Figure 2c) and total mean GP values (Figure 2d).

### 3.3. Cytoskeletal Abnormality in MNPs@SiO_2_(RITC)-Treated hBM-MSCs

We investigated the changes in cell morphology and focal adhesion in MNPs@SiO_2_(RITC)-treated hBM-MSCs. As shown in Figure 2, cell shrinkage was observed with reduction in the attached area of MNPs@SiO_2_(RITC)-treated hBM-MSCs, indicating cytoskeletal changes and abnormal adhesion, which are followed by reduced membrane fluidity due to lipid peroxidation. Furthermore, immunocytochemical analysis of β-tubulin, a major cytoskeletal protein, showed shrinkage of the cell and β-tubulin structure, although interaction between MNPs@SiO_2_(RITC) and β-tubulin was not detected (Figure 3).

Immunocytochemical analysis of F-actin, a major cytoskeletal protein, co-stained with vinculin, a focal adhesion marker, showed that unlike that in non-treated control hBM-MSCs, lamellipodia and filopodia disappeared, and vinculin was abnormally congregated in the central region of hBM-MSCs treated with 1.0 µg/µL MNPs@SiO_2_(RITC) (Figure 4a). However, specific interaction between MNPs@SiO_2_(RITC) and F-actin or vinculin was not detected. Furthermore, compared to that in the non-treated control, the attached relative cell area was reduced in 0.1 and 1.0 µg/µL MNPs@SiO_2_(RITC)-treated hBM-MSCs (Figure 4b). Levels of phosphorylated proto-oncogene tyrosine-protein kinase SRC (c-SRC) and focal adhesion kinase (FAK), which are activated forms of the focal adhesion proteins SRC and FAK, respectively, were reduced in cells treated with 1.0 µg/µL MNPs@SiO_2_(RITC) (Figure 4c).

### 3.4. Reduction in Traction Force of hBM-MSCs after MNPs@SiO_2_(RITC) Treatment

To analyze, in detail, the parameters of cell adhesion, cell spread areas were measured using images of MNPs@SiO_2_(RITC)-treated hBM-MSCs and submicron pillars at 12 h after cell seeding (Figure 5a). Compared to that in the non-treated control cells, cell spreading of hBM-MSCs treated with 0.1 and 1.0 µg/µL MNPs@SiO_2_(RITC) decreased (Figure 5b). Furthermore, the spread areas of hBM-MSCs treated with 0.1 and 1.0 µg/µL MNPs@SiO_2_(RITC) were significantly smaller than those of the non-treated control cells (Figure 5c).

The pillar deflection in the magnified images was used to measure pillar displacement (Figure 5d) and calculate traction forces (Figure 5e,f). To calculate the traction force of a pillar, displacement of each pillar was multiplied by the bending stiffness of the pillar [43]. There were no significant changes in pillar displacement and the average traction force of MNPs@SiO_2_(RITC)-treated hBM-MSCs. However, the total traction force of hBM-MSCs treated with 1.0 µg/µL MNPs@SiO_2_(RITC) was significantly lower than those of hBM-MSCs treated with 0.1 µg/µL MNPs@SiO_2_(RITC) and non-treated control hBM-MSCs. The results showed that cell attachment was impaired after treatment with 1.0 µg/µL MNPs@SiO_2_(RITC). Taken together with the spread area and traction force data, this result implies that the decrease in total traction force of hBM-MSCs results from cell shrinkage upon MNPs@SiO_2_(RITC) treatment.

### 3.5. Reduction in Migratory Activity of hBM-MSCs after MNPs@SiO_2_(RITC) Treatment

To evaluate membrane fluidity and adhesion-related biological functions, we assessed the effect of treatment with 0.1 or 1.0 µg/µL MNPs@SiO_2_(RITC) on the migratory activity of hBM-MSCs using wound healing and invasion assays. Compared to those of non-treated controls and cells treated with 0.1 μg/μL MNPs@SiO_2_(RITC), the migratory ability of hBM-MSCs treated with 1.0 μg/μL MNPs@SiO_2_(RITC) was seen to be significantly impaired through the wound healing assay (Figure 6a). Results of the invasion assay showed that compared to that of the non-treated controls, the invasion ability of hBM-MSCs was significantly impaired by MNPs@SiO_2_(RITC) treatment in a dose-dependent manner (Figure 6b).

## 4. Discussion

This study used molecular cellular biology tests, TIRFM measurement of membrane fluidity, micropillar measurement of traction force, and invasion and migration analyses to evaluate the biophysical effects of MNPs@SiO_2_(RITC) treatment in hBM-MSCs. Our results indicated that MNPs@SiO_2_(RITC) usage should be minimized in cell labelling to preserve the biophysical effect of hBM-MSCs.

Reduction in cell viability and ROS generation have been reported in MSCs treated with nanoparticles of various types and sizes [58,59,60]. In this study, cell viability decreased slightly by about 10% in hBM-MSCs treated with both 0.1 and 1.0 µg/µL MNPs@SiO_2_(RITC) and silica nanoparticles. This result is consistent with that of a previous study on 0–1.0 µg/µL MNPs@SiO_2_(RITC)-treated human cord blood–derived MSCs [48]. However, the viability of HEK293 cells treated with MNPs@SiO_2_(RITC) has been shown to decrease by about 1%–3% [24]. These discrepancies may be related to the cell-specific characteristics, as MSCs are more sensitive to excessive ROS induced by nanoparticles than differentiated cell lines [61].

We analyzed the biophysical changes and membrane fluidity in MNPs@SiO_2_(RITC)-treated hBM-MSCs, the effects of lipid peroxidation by MNPs@SiO_2_(RITC)-induced ROS, and the physical interaction between the cell membrane and MNPs@SiO_2_(RITC). Our results suggested that cell membrane damage induced by MNPs@SiO_2_(RITC) can occur via direct interaction between membrane lipids and nanoparticles [62] and that lipid peroxidation in nanoparticle-treated hBM-MSCs can induce oxidative membrane damage, resulting in biological alterations [63,64]. In addition, the biological effects of MNPs@SiO_2_(RITC) were caused by their silica shell rather than the cobalt ferrite core compounds, as reported previously [23,24].

We also observed that MNPs@SiO_2_(RITC)-treated hBM-MSCs showed a rounded and shrunken morphology with disrupted cytoskeletal structure. However, specific interactions between MNPs@SiO_2_(RITC) and actin or tubulin were not observed. Generally, the cytoskeleton is tightly linked to the membrane via phosphoinositides and linker proteins, such as spectrin, Ezrin/radixin/moesin (ERM), and myosin-I [33,34,65]. Thus, biophysical changes in the membrane are reflected as cytoskeletal changes in hMSCs [65]. Based on the relationship between the cytoskeleton and ROS, high ROS levels have been shown to induce microtubule dysfunction and sever F-actin structure [35]. Furthermore, endocytosis is another potential mechanism underlying cytoskeletal changes. During endocytosis, cells undergo reorganization of the cytoskeleton and the membrane [66]. Thus, the cytoskeletal rearrangement may be caused by ROS generation and bulk endocytosis post-MNPs@SiO_2_(RITC) treatment.

Previous studies have suggested that the major mechanism underlying nanoparticle-induced cell shrinkage and abnormal formation of focal adhesions involves cytoskeletal depolymerization and an increase in cellular traction force [67,68,69]. We observed that the lamellipodia and filopodia structures of MNPs@SiO_2_(RITC)-treated hBM-MSCs were disrupted and the phosphorylation of FAK and c-SRC, which are markers of focal adhesion formation, was reduced. However, there were no changes in the traction forces, although the total traction force was reduced. These results indicated that MNPs@SiO_2_(RITC)-induced reduction of FAK and c-SRC phosphorylation corresponded with the reduction in the attached area of cells and the number of focal adhesions during cell shrinkage and cytoskeletal structure disruption after MNPs@SiO_2_(RITC) treatment.

We observed that the changes in biophysical properties, such as reduced membrane fluidity, cell shrinkage with disrupted cytoskeletal structure, and reduced total traction force in MNPs@SiO_2_(RITC)-treated hBM-MSCs, were tightly linked to each other and contributed to the reduction in the migratory activity of hBM-MSCs. These findings are supported by two previous findings: (i) Actin assembly-based protrusions and generation of traction forces are key processes in cell migration [70]. Furthermore, migratory activity is highly related to cellular biophysical properties, such as membrane fluidity and traction force [71,72]. (ii) Condensation of the cytoskeleton due to MNPs@SiO_2_(RITC) treatment increases cell stiffness, which impedes cellular movement in the contractile machinery (shrinkage) [71].

Evaluation of the migratory activity is a major requirement for hBM-MSC applications and for successful outcome of stem cell therapy and tracking studies [66]. In addition, appropriate localization of hBM-MSCs in damaged tissues is important for the therapeutic effect of trophic factors and cytokines secreted by hBM-MSCs [73]. Previously, we have showed that PKH-26, a red fluorescent cell labelling dye, labelled hBM-MSCs in a localized, spotted-like pattern, implying secretion of the red dye outside the cell, in the border region of a lesion in a rat model of ischemic stroke [50]. However, MNPs@SiO_2_(RITC)-labelled human umbilical cord blood–derived MSCs in vivo rarely appeared as spotted particles in a mouse model, suggesting retention of MNPs@SiO_2_(RITC) in the cell for good tracking efficacy [48]. Hence, we believe that MNPs@SiO_2_(RITC) is better than the PKH-26 dye for hBM-MSC tracing. Further studies are required (i.e., in vivo experiment) for tracking MNPs@SiO_2_(RITC)-treated hBM-MSCs.

Based on our analysis of the effect of MNPs@SiO_2_(RITC) treatment on the biophysical and biological functions of hBM-MSCs, we suggest the mechanism of action of MNPs@SiO_2_(RITC) as follows: (i) MNPs@SiO_2_(RITC) are internalized into hBM-MSCs; (ii) intracellular ROS are generated by the internalized MNPs@SiO_2_(RITC); (iii) the ROS induces peroxidation of lipids, which are major components of the cell membrane; (iv) peroxidation decreases membrane fluidity; (v) cell shrinkage is induced by the reduction in membrane fluidity; (vi) owing to the cell shrinkage, abnormality in adhesion and reduction in total traction force are induced; (vii) owing to the reduction in membrane fluidity and abnormality in adhesion, fundamentally induced by intracellular ROS, the migratory activity of MNPs@SiO_2_(RITC)-treated hBM-MSCs decreases. The effect of this treatment on other cell functions will be addressed in future investigations.

In conclusion, our findings suggest that high-dose MNPs@SiO_2_(RITC) can alter biophysical properties and reduce the migratory activity of MNPs@SiO_2_(RITC)-treated hBM-MSCs. Thus, nanoparticles used for stem cell trafficking or clinical applications should be labelled using optimal nanoparticle concentrations to preserve hBM-MSC migratory activity and ensure successful outcomes following stem cell localization.

## Figures and Tables

**Figure 1 nanomaterials-09-01475-f001:**
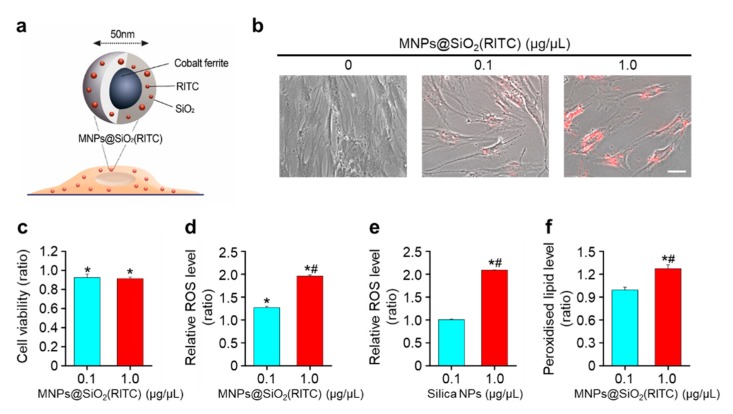
Intracellular reactive oxygen species (ROS) generation and lipid peroxidation in MNPs@SiO_2_(RITC)-treated hBM-MSCs. (**a**) Schematic showing MNPs@SiO_2_(RITC) composition. (**b**) Morphological analysis of non-treated control and MNPs@SiO_2_(RITC)-treated hBM-MSCs. Scale bar = 50 µm (**c**) Cell viability assay with hBM-MSCs treated with MNPs@SiO_2_(RITC) for 12 h. Evaluation of intracellular ROS generation using DCFH-DA for 12 h in HEK293 cells treated with (**d**) MNPs@SiO_2_(RITC) and (**e**) silica NPs. The non-oxidized DCFH-DA was used as the blank. (**f**) Evaluation of peroxidized lipids using ferrous thiocyanate. Ferrous thiocyanate was used as the blank. Data represent mean ± SD of three independent experiments. * *p* < 0.05 vs. non-treated control, ^#^
*p* < 0.05 for the comparison between 0.1 and 1.0 µg/µL MNPs@SiO_2_(RITC) or silica NP-treated cells. Data represent mean ± SD of three independent experiments.

**Figure 2 nanomaterials-09-01475-f002:**
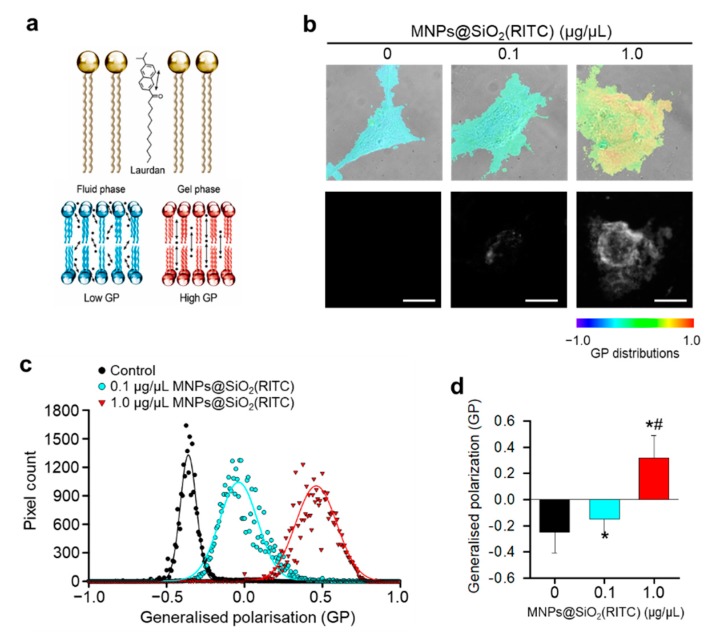
Laurdan generalized polarizations (GP) images and GP frequency distributions of hBM-MSCs treated with MNPs@SiO_2_(RITC) for 12 h. (**a**) Schematic of Laurdan for measuring membrane GP value. (**b**) Merged differential interference contrast (DIC) and total internal reflection fluorescence microscopy (TIRFM) images of human bone marrow-derived mesenchymal stem cells (hBM-MSCs) in each upper panel. Distributions of magnetic nanoparticles incorporating rhodamine B isothiocyanate (MNPs@SiO_2_(RITC)) are indicated in each lower panel. GP distributions ranged from −1.0 to 1.0. Scale bar = 2.5 µm. (**c**) GP frequency distributions of cells. GP values of each pixel are represented as dots and were fitted to Gaussian distributions. (**d**) Total GP values. Data represent mean ± SD of three independent experiments (*n* = 10). * *p <* 0.05 vs. non-treated control, ^#^
*p <* 0.05 for the comparison between cells treated with 0.1 and 1.0 µg/µL of MNPs@SiO_2_(RITC).

**Figure 3 nanomaterials-09-01475-f003:**
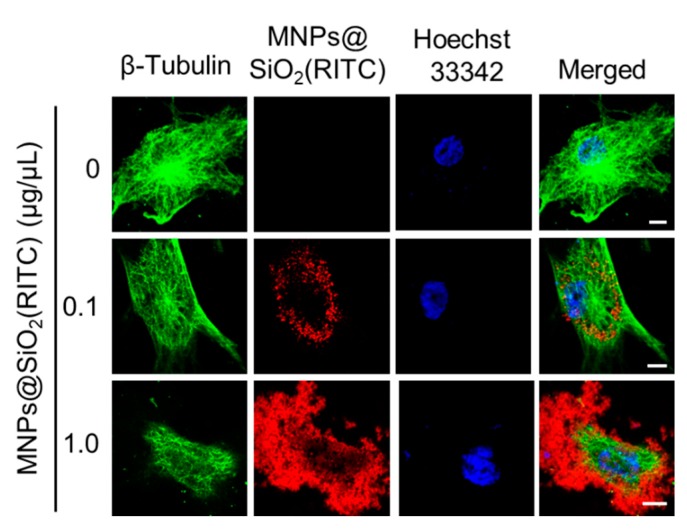
Tubulin-based analysis of the cytoskeleton of hBM-MSCs treated with MNPs@SiO_2_(RITC) for 12 h. β-Tubulin stained images of hBM-MSCs treated with MNPs@SiO_2_(RITC). Blue, Hoechst 33342; Green, β-tubulin; red, MNPs@SiO_2_(RITC). Scale bar = 10 μm.

**Figure 4 nanomaterials-09-01475-f004:**
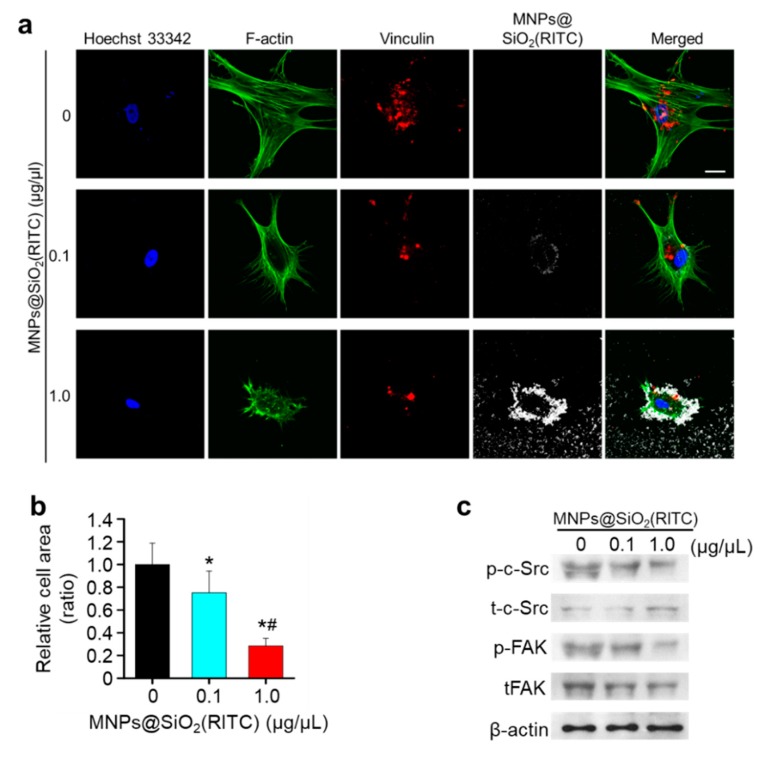
Shrinkage and abnormal focal adhesion of hBM-MSCs treated with MNPs@SiO_2_(RITC). (**a**) Images of F-actin based cytoskeleton and focal adhesion in MNPs@SiO_2_(RITC)-treated HEK293 cells. Green, F-actin; red, vinculin; white, MNPs@SiO_2_(RITC); blue, Hoechst 33342. Scale bar = 20 µm. (**b**) Relative attached area of MNPs@SiO_2_(RITC)-treated hBM-MSCs compared to non-treated control. Data represent mean ± SD (*n* > 30). * *p <* 0.05 vs. non-treated control. ^#^
*p* < 0.05 for the comparison between 0.1 and 1.0 µg/µL MNPs@SiO_2_(RITC)-treated cells. (**c**) Immunoblotting analysis associated with focal adhesion. p-, phosphorylated protein; t, total protein. β-Actin was used as an internal control.

**Figure 5 nanomaterials-09-01475-f005:**
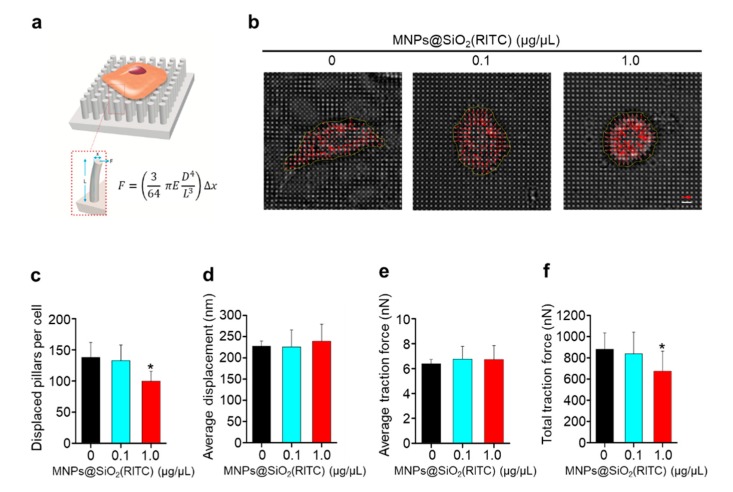
Change in pillar deflection, traction force, aspect ratio, and surface area of hBM-MSCs treated with MNPs@SiO_2_(RITC). (**a**) Schematic of traction force measurement using a micropillar. *F* = traction force; *E* = Young modulus of the pillar; *D* = diameter of pillar; *L* = length of pillar; Δ*x* = pillar displacement. (**b**) Representative images showing the concentration of MNPs@SiO_2_(RITC) inside the cell, pillar deflections, and magnified pillar deflections at the edge of the cell (left to right). The red arrow represents 356 nm of deflection and the white bar represents 8 µm. The yellow line indicates the approximate cell boundary. The direction and length of the red arrow indicate the magnitude and direction of pillar deflection, respectively. (**c**) Displaced pillar number of pillar array (**d**) Average displacement of each pillar under the cell. (**e**) Average traction force of each pillar under the cell and (**f**) total traction force of pillars beneath the cell. Data represent mean ± SD (*n* = 21). * *p* < 0.05 vs. non-treated control.

**Figure 6 nanomaterials-09-01475-f006:**
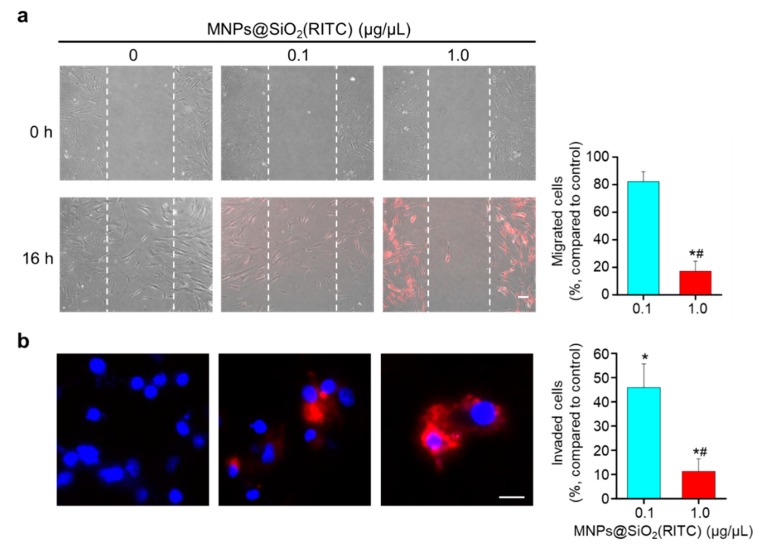
Migratory activity of hBM-MSCs treated with MNPs@SiO_2_(RITC). (**a**) Representative images of wound healing assay and quantitative image analysis. Images of the initial wounded (0 h) layer are shown the upper panels. Images of cells after MNPs@SiO_2_(RITC) treatment for 16 h are shown in the lower panels. Scale bar = 100 μm. Quantitative image analysis of migrated cells in MNPs@SiO_2_(RITC)-treated hBM-MSCs are shown in the bar graph. (**b**) Representative images of hBM-MSCs and quantitative image analysis of the invasion assay results after MNPs@SiO_2_(RITC) treatment for 12 h. Red, MNPs@SiO_2_(RITC); blue, Hoechst 33342. Scale bar = 20 μm. Quantitative image analysis of invaded cells in MNPs@SiO_2_(RITC)-treated hBM-MSCs are shown in the bar graph. Data represent mean ± SD of three independent experiments. Data represent mean ± SD of three independent experiments. * *p <* 0.05 vs. non-treated control, ^#^
*p* < 0.05 for the comparison between cells treated with 0.1 and 1.0 µg/µL of MNPs@SiO_2_(RITC).

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
