# Peer review of "Silica-Coated Magnetic Nanoparticles Decrease Human Bone Marrow-Derived Mesenchymal Stem Cell Migratory Activity by Reducing Membrane Fluidity and Impairing Focal Adhesion"

_nanomaterials, 2019, doi:10.3390/nano9101475_

Round 1
Reviewer 1 Report
Abstract: abstract is written rather poorly and need better structure with clear knowledge gap, aim, objective followed by results and brief implications of the results.
While the paper describes very well the effect of NPs on cell, there is lack of data on the total amount of NPs internalised by cell. Nanodosimetry experiments were not done/presented. The amount will eventually determine the cellular repossess and it would be advised to have this data or at least comment on it.
Did you establish IC50 for the NPs?
Also I would suggest to name subsections not by technique but rather by parameter that you determine with specific test e.g. not MTS but rather Cell metabolic activity assay.
You have to say more about pillar structures in the text, what did you do that, what was the purpose?
Also some of the results sections refer to previous papers and provide comments. I would remove this and only keep results and move commentary to discussion section; e.g. ‘A previous study revealed that MNPs@SiO2(RITC) induced intracellular ROS generation 238 through mitochondrial dysfunction, however, impaired viability was not detected in human 239 embryonic kidney 293 (HEK 293) cells [24].’
Discussion is good, however I would strongly suggest to add section with bullet points which clearly explain the effect of NPs on cell function.
Author Response
Answers to the comments
Reviewer # 1
Abstract: abstract is written rather poorly and need better structure with clear knowledge gap, aim, objective followed by results and brief implications of the results.
à As you have pointed out, we modified the abstract in page 1 as
Abstract: For stem cell-based therapies, the fate and distribution of stem cells should be traced using non-invasive or histological methods and a nanomaterial-based labelling agent. However, evaluation of the biophysical effects and related biological functions of nanomaterials in stem cells remains challenging. Here, we aimed to investigate the biophysical effects of nanomaterials on stem cells, including those on membrane fluidity, using total internal reflection fluorescence microscopy, and traction force, using micropillars of human bone marrow-derived mesenchymal stem cells (hBM-MSCs) labelled with silica-coated magnetic nanoparticles incorporating rhodamine B isothiocyanate (MNPs@SiO2(RITC)). Furthermore, to evaluate the biological functions related to these biophysical changes, we assessed the cell viability, reactive oxygen species (ROS) generation, intracellular cytoskeleton, and the migratory activity of MNPs@SiO2(RITC)-treated hBM-MSCs. Compared to that in the control, cell viability decreased by 10% and intracellular ROS increased by 2-fold due to the induction of 20% higher peroxidised lipid in hBM-MSCs treated with 1.0 µg/µL MNPs@SiO2(RITC). Membrane fluidity was reduced by MNPs@SiO2(RITC)-induced lipid oxidation in a concentration-dependent manner. In addition, cell shrinkage with abnormal formation of focal adhesions and ~30% decreased total traction force were observed in cells treated with 1.0 µg/µL MNPs@SiO2(RITC) without specific interaction between MNPs@SiO2(RITC) and cytoskeletal proteins. Furthermore, the migratory activity of hBM-MSCs, which was highly related to membrane fluidity and cytoskeletal abnormality, decreased significantly after MNPs@SiO2(RITC) treatment. These observations indicated that the migratory activity of hBM-MSCs was impaired by MNPs@SiO2(RITC) treatment due to changes in stem-cell biophysical properties and related biological functions, highlighting the important mechanisms via which nanoparticles impair migration of hBM-MSCs. Our findings indicate that nanoparticles used for stem cell trafficking or clinical applications should be labelled using optimal nanoparticle concentrations to preserve hBM-MSC migratory activity and ensure successful outcomes following stem cell localisation.
While the paper describes very well the effect of NPs on cell, there is lack of data on the total amount of NPs internalised by cell. Nanodosimetry experiments were not done/presented. The amount will eventually determine the cellular repossess and it would be advised to have this data or at least comment on it.
à The amount of internalised MNPs@SiO2(RITC) were not presented in this study. A previous study determined that the MCF-7 cell line internalised ~ 105 particles of MNPs@SiO2(RITC) per cell using inductively coupled plasma atomic emission spectrometry. Moreover, our previous study determined fluorescence intensity by treating HEK293 cells with MNPs@SiO2(RITC) at concentrations ranging from 0.01 to 2.0 μg/μl for 12 h and calculating the uptake efficiency at each concentration. In this study, the fluorescence intensity of MNPs@SiO2(RITC)-labelled hBM-MSCs were similar with the MNPs@SiO2(RITC)-labelled HEK293 cells, we assumed that the amount of internalised MNPs@SiO2(RITC) is similar with previous study. We added this information in page 2 as
2.1. MNPs@SiO2(RITC) and silica nanoparticles (NPs)
MNPs@SiO2(RITC) particles, composed of a ~9 nm cobalt ferrite core (CoFe2O3) chemically bonded to rhodamine isothiocyanate dye (RITC) and coated by a silica shell [4], were purchased from BITERIALS (Seoul, South Korea). Previously, these nanoparticles have been characterized for confirming their quality [45]. Size distribution and morphology are important factors determining the uniformity of nanoparticles and were analysed using electron and atomic microscopy [45]. Hydrodynamic size, polydispersity, and surface charge were determined using dynamic light scattering [46]. The purity and contents of nanoparticles are usually analysed using an X-ray based technique [45]. In this study, X-ray diffraction (XRD) analysis using a high power X-Ray diffractometer (Ultima III, Rigaku, Japan) confirmed the structure of MNPs@SiO2(RITC) (data not shown). The silica NPs were composed of identical materials and were of a similar size as the MNPs@SiO2(RITC) shell, and their biological effects were similar to those of MNPs@SiO2(RITC) [23,24,47,48]. The diameters of the MNPs@SiO2(RITC) and silica NPs were 50 nm, and the zeta potential of MNPs@SiO2(RITC) was between -40 to -30 mV [4,47]. A previous study determined ~ 105 particles of MNPs@SiO2(RITC) per cell in MNPs@SiO2(RITC)-treated MCF-7 cells using inductively coupled plasma atomic emission spectrometry [4]. Furthermore, in previous reports, the dosage was determined by measuring the fluorescence intensity of HEK293 cells treated with MNPs@SiO2(RITC) at concentrations ranging from 0.01 to 2.0 μg/μL for 12 h. The optimal concentration of MNPs@SiO2(RITC) was 0.1 µg/µL for in vitro use, whereas 1.0 µg/µL was the plateau concentration for cellular uptake [24]. Furthermore, MNPs@SiO2(RITC) concentrations ranging from 0 to 1.0 μg/μL have been used for MRI contrasting without toxicological effects on human cord blood-derived MSCs [49], and caused changes in gene expression and metabolic profiles similar to those of the control HEK293 cells at 0.1 µg/µL [24]. In addition, the uptake efficiency of MNPs@SiO2(RITC) almost plateaued at 1.0 µg/µL in HEK293 cells [24,25]. The dose-dependent fluorescence intensity of MNPs@SiO2(RITC)-labelled hBM-MSCs was similar to those of labelled HEK293 cells. In addition, the viability of human cord blood-derived MSCs was determined to assess the cytotoxic effect of MNPs@SiO2(RITC) after 24, 48, and 72 h of treatment with 0–1.0 µg/µL MNPs@SiO2(RITC); compared to the control group, no significant cytotoxic effect was observed [49]. Therefore, in this study, hBM-MSCs were treated with 0.1 µg/µL (low dose) MNPs@SiO2(RITC)or 1.0 µg/µL (high dose), similarly to previous reports [23,24,48].
Did you establish IC50 for the NPs?
à We did not establish IC50 for MNPs@SiO2(RITC) because previous study reported that viability changes are not measurable in 0 to 1.0 μg/μl MNPs@SiO2(RITC) treated MSCs. Moreover, over 1.0 µg/µL of MNPs@SiO2(RITC) is plateau concentration for cellular uptake based on previous study. We added this information in page 2 as
2.1. MNPs@SiO2(RITC) and silica nanoparticles (NPs)
MNPs@SiO2(RITC) particles, composed of a ~9 nm cobalt ferrite core (CoFe2O3) chemically bonded to rhodamine isothiocyanate dye (RITC) and coated by a silica shell [4], were purchased from BITERIALS (Seoul, South Korea). Previously, these nanoparticles have been characterized for confirming their quality [45]. Size distribution and morphology are important factors determining the uniformity of nanoparticles and were analysed using electron and atomic microscopy [45]. Hydrodynamic size, polydispersity, and surface charge were determined using dynamic light scattering [46]. The purity and contents of nanoparticles are usually analysed using an X-ray based technique [45]. In this study, X-ray diffraction (XRD) analysis using a high power X-Ray diffractometer (Ultima III, Rigaku, Japan) confirmed the structure of MNPs@SiO2(RITC) (data not shown). The silica NPs were composed of identical materials and were of a similar size as the MNPs@SiO2(RITC) shell, and their biological effects were similar to those of MNPs@SiO2(RITC) [23,24,47,48]. The diameters of the MNPs@SiO2(RITC) and silica NPs were 50 nm, and the zeta potential of MNPs@SiO2(RITC) was between -40 to -30 mV [4,47]. A previous study determined ~ 105 particles of MNPs@SiO2(RITC) per cell in MNPs@SiO2(RITC)-treated MCF-7 cells using inductively coupled plasma atomic emission spectrometry [4]. Furthermore, in previous reports, the dosage was determined by measuring the fluorescence intensity of HEK293 cells treated with MNPs@SiO2(RITC) at concentrations ranging from 0.01 to 2.0 μg/μL for 12 h. The optimal concentration of MNPs@SiO2(RITC) was 0.1 µg/µL for in vitro use, whereas 1.0 µg/µL was the plateau concentration for cellular uptake [24]. Furthermore, MNPs@SiO2(RITC) concentrations ranging from 0 to 1.0 μg/μL have been used for MRI contrasting without toxicological effects on human cord blood-derived MSCs [49], and caused changes in gene expression and metabolic profiles similar to those of the control HEK293 cells at 0.1 µg/µL [24]. In addition, the uptake efficiency of MNPs@SiO2(RITC) almost plateaued at 1.0 µg/µL in HEK293 cells [24,25]. The dose-dependent fluorescence intensity of MNPs@SiO2(RITC)-labelled hBM-MSCs was similar to those of labelled HEK293 cells. In addition, the viability of human cord blood-derived MSCs was determined to assess the cytotoxic effect of MNPs@SiO2(RITC) after 24, 48, and 72 h of treatment with 0–1.0 µg/µL MNPs@SiO2(RITC); compared to the control group, no significant cytotoxic effect was observed [49]. Therefore, in this study, hBM-MSCs were treated with 0.1 µg/µL (low dose) MNPs@SiO2(RITC)or 1.0 µg/µL (high dose), similarly to previous reports [23,24,48].
Also I would suggest to name subsections not by technique but rather by parameter that you determine with specific test e.g. not MTS but rather Cell metabolic activity assay.
à We changed the terminology as your recommendation.
2.4. Cell viability assay
For analysis of cell viability, the CellTiter 96-cell proliferation assay kit (MTS, Promega, USA) was used, according to the manufacturer’s instructions. Briefly, 2 × 104 hBM-MSCs were seeded on 96-well assay plates. After 16 h, the hBM-MSCs were washed with PBS and treated with MNPs@SiO2(RITC) for 12 h. The hBM-MSCs were then washed with PBS to remove excess MNPs@SiO2(RITC), and MTS solution was added to each well (1/10 volume of media). Subsequently, the plate was incubated for 1 h in a 5% CO2 chamber maintained at 37°C. The absorbance of the soluble formazan was measured using a plate reader (Molecular Devices, USA) at 490 nm. Values were normalised relative to the protein absorbance value for each corresponding group.
2.13. Invasion assay
Invasion assays of hBM-MSCs were performed using an 8-µm pore size transwell polycarbonate membrane (Corning, CA, USA). The upper side of the membrane was coated with Matrigel (1:10 dilution in 0.01 M Tris pH 8.0, 0.7% NaCl) for 2 h at 37°C. hBM-MSCs were treated with MNPs@SiO2(RITC) for 12 h. Next, 2.5 × 104 hBM-MSCs were transferred to the upper chamber of the transwell in serum-free media, and 10% FBS containing medium was added to the lower chamber as a chemoattractant. The cells were incubated for 12 h at 37°C. Subsequently, the cells on the upper side of the membrane were removed with a cotton swab, and the invading cells on the lower side of the membrane were fixed in Cytofix buffer (BD, San Jose, CA, USA) and stained with Hoechst 33342. Images were acquired using an Axio Vert 200M fluorescence microscope (Zeiss, Jena, Germany). The excitation wavelengths for MNPs@SiO2(RITC) and Hoechst 33342 were 530 nm and 405 nm, respectively. The number of invading cells was counted using the ImageJ software.
You have to say more about pillar structures in the text, what did you do that, what was the purpose?
à As you have pointed out, we added the information of pillar structures in ‘Introduction’ and ‘Materials and Methods’ section
The focal adhesion of hBM-MSCs is strongly associated with changes in cellular traction forces [41]. Elastomeric pillar arrays are considered excellent for measuring cellular traction force by calculating the nanometric level of pillar deflection [42,43]. In addition, sub-micron pillar arrays have been shown to mimic continuous substrates of specific rigidity [44]. Thus, biophysical changes in nanoparticle-treated cells have been quantitatively studied using elastomeric submicron pillars [42,44].
2.10. Microfabrication of pillar arrays
A standard photolithograph was used to fabricate a mould with arrays of holes over a silicon wafer [56]. To fabricate pillar arrays, Polydimethylsiloxane (PDMS) was mixed at 10:1 ratio with its curing agent (Sylgard 184; Dow Corning) and degassed for 15 min. Next, it was spin-coated over the mould, degassed again for 30 min to remove trapped air bubbles within the mixture, and cured at 80°C for 4 h and 30 min until the Young modulus of the PDMS reached 2 ± 0.1 MPa. Subsequently, the cured pillar array was carefully removed from the mould. In the case of micron scale pillars, cellular contractions occurred around individual pillars (diameter 2 μm) [44,57]. Thus, we generated linearly arranged pillar arrays of 900 nm diameter, 1 µm height, and a pillar diameter two times the centre-to-centre distance between pillars. The bending stiffness of the pillar was calculated based on Euler-Bernoulli beam theory:
Where D is the diameter, L is the length, and E is the Young modulus of the pillar [44]. The bending stiffness (k) of pillar arrays used in this study was 28.8 nN/µm.
Additional references
Tan, J.L.; Tien, J.; Pirone, D.M.; Gray, D.S.; Bhadriraju, K.; Chen, C.S. Cells lying on a bed of microneedles: an approach to isolate mechanical force. Proc Natl Acad Sci U S A 2003, 100, 1484-1489, doi:10.1073/pnas.0235407100.
Cui, Y.; Hameed, F.M.; Yang, B.; Lee, K.; Pan, C.Q.; Park, S.; Sheetz, M. Cyclic stretching of soft substrates induces spreading and growth. Nat Commun 2015, 6, 6333, doi:10.1038/ncomms7333.
Also some of the results sections refer to previous papers and provide comments. I would remove this and only keep results and move commentary to discussion section; e.g. ‘A previous study revealed that MNPs@SiO2(RITC) induced intracellular ROS generation 238 through mitochondrial dysfunction, however, impaired viability was not detected in human 239 embryonic kidney 293 (HEK293) cells [24].’
à We changed the results sections as your recommendation.
3.1. Decrease in cell viability and ROS generation of MNPs@SiO2(RITC)-treated hBM-MSCs
A previous study revealed that MNPs@SiO2(RITC) induced intracellular ROS generation through mitochondrial dysfunction; however, impaired viability was not detected in human embryonic kidney 293 (HEK293) cells [24]. In this study, To evaluate the viability of and ROS generation in MNPs@SiO2(RITC)-treated hBM-MSCs, hBM-MSCs were treated with MNPs@SiO2(RITC) for 12 h before analysis (Fig. 1a). A monolayer of hBM-MSCs was clearly observed for the non-treated control cells, while the monolayer was disintegrated for the MNPs@SiO2(RITC)-treated hBM-MSCs (Fig. 1b). Furthermore, compared to that in the non-treated control, the viability of hBM-MSCs decreased by ~10% upon treatment with 0.1 and 1.0 µg/µL MNPs@SiO2(RITC) (Fig. 1c). However, the viability of cells treated with 0.1 µg/µL MNPs@SiO2(RITC) was not statistically significantly different from that of cells treated with 1.0 µg/µL MNPs@SiO2(RITC). Intracellular ROS generation was evaluated using 2’,7’-dichlorodihydrofluorescin diacetate (DCFH-DA) staining in MNPs@SiO2(RITC)-treated hBM-MSCs. Treatment with MNPs@SiO2(RITC) and 50 nm-sized silica NPs increased intracellular ROS levels (Fig. 1d, e). In particular, compared to that in non-treated control cells and cells treated with 0.1 µg/µL NPs, intracellular ROS levels increased by more than 50% in cells treated with 1.0 µg/µL MNPs@SiO2(RITC) and silica-NPs. Furthermore, the level of ROS-induced peroxidised lipids increased by ~30% in hBM-MSCs treated with 1.0 µg/µL MNPs@SiO2(RITC) (Fig. 1f).
3.2. Reduction in membrane fluidity of hBM-MSCs after MNPs@SiO2(RITC) treatment
It was postulated that biophysical changes caused by nanoparticles initially occurs in the cell membrane because they first cross the membrane via endocytosis to induce ROS generation [51,52]. Thus, To analyse changes in membrane fluidity due to peroxidation of lipids, the MNPs@SiO2(RITC)-treated hBM-MSCs were stained with Laurdan and generalised polarisation (GP) values were calculated using TIRFM (Fig. 2a). The number of high-GP areas on the hBM-MSC surface—corresponding to rigid domains—increased upon MNPs@SiO2(RITC) treatment. In particular, the abundantly distributed region of MNPs@SiO2(RITC) majorly co-localised with the high GP-distribution region in a GP scale of −1.0 to 1.0 (Fig. 2b). GP frequency distribution values of MNPs@SiO2(RITC) treated-hBM-MSCs were subtracted from the corresponding values of the non-treated control hBM-MSCs to obtain frequency difference curves (Fig. 2c) and total mean GP values (Fig. 2d).
3.3. Cytoskeletal abnormality in MNPs@SiO2(RITC)-treated hBM-MSCs
Membrane fluidity is closely related to cell morphology and focal adhesion, because the membrane and cytoskeleton are tightly associated with phosphoinositides and linker proteins [33,34,53]. Therefore, We investigated the changes in cell morphology and focal adhesion in MNPs@SiO2(RITC)-treated hBM-MSCs. As shown in Fig. 2, cell shrinkage was observed with reduction in the attached area of MNPs@SiO2(RITC)-treated hBM-MSCs, indicating cytoskeletal changes and abnormal adhesion, which are followed by reduced membrane fluidity due to lipid peroxidation. Furthermore, immunocytochemical analysis of β-tubulin, a major cytoskeletal protein, showed shrinkage of the cell and β-tubulin structure, although interaction between MNPs@SiO2(RITC) and β-tubulin was not detected (Fig. 3).
3.4. Reduction in traction force of hBM-MSCs after MNPs@SiO2(RITC) treatment
Based on the reduced phosphorylation of c-Src and FAK, we next evaluated the effects of MNPs@SiO2(RITC) on cell polarity and spreading, as local contraction of hBM-MSCs is tightly associated with cell polarity, spreading, and changes in focal adhesions [54]. Additionally, To analyse, in detail, the parameters of cell adhesion, cell spread areas were measured using images of MNPs@SiO2(RITC)-treated hBM-MSCs and submicron pillars at 12 h after cell seeding (Fig. 5a). Compared to that in the non-treated control cells, cell spreading of hBM-MSCs treated with 0.1 and 1.0 µg/µL MNPs@SiO2(RITC) decreased (Fig. 5b). Furthermore, the spread areas of hBM-MSCs treated with 0.1 and 1.0 µg/µL MNPs@SiO2(RITC) were significantly smaller than those of the non-treated control cells (Fig. 5c).
3.5. Reduction in migratory activity of hBM-MSCs after MNPs@SiO2(RITC) treatment
Subsequently, we investigated the biophysical properties of hBM-MSCs altered by MNPs@SiO2(RITC), including membrane fluidity and traction force, which are tightly linked and highly related with migratory activity of cells [55,56]. To evaluate membrane fluidity and adhesion-related biological functions, we assessed the effect of treatment with 0.1 or 1.0 µg/µL MNPs@SiO2(RITC) on the migratory activity of hBM-MSCs using wound healing and invasion assays. Compared to those of non-treated controls and cells treated with 0.1 μg/μL MNPs@SiO2(RITC), the migratory ability of hBM-MSCs treated with 1.0 μg/μL MNPs@SiO2(RITC) was seen to be significantly impaired through the wound healing assay (Fig. 6a). Results of the invasion assay showed that compared to that of the non-treated controls, the invasion ability of hBM-MSCs was significantly impaired by MNPs@SiO2(RITC) treatment in a dose-dependent manner (Fig. 6b).
Discussion is good, however I would strongly suggest to add section with bullet points which clearly explain the effect of NPs on cell function.
à We added the discussion section for explanation of the effect of MNPs@SiO2(RITC) on cell function as your recommendation.
Based on our analysis of the effect of MNPs@SiO2(RITC) treatment on the biophysical and biological functions of hBM-MSCs, we suggest the mechanism of action of MNPs@SiO2(RITC) as follows: (i) MNPs@SiO2(RITC) are internalised into hBM-MSCs; (ii) intracellular ROS are generated by the internalised MNPs@SiO2(RITC); (iii) the ROS induces peroxidation of lipids, which are major components of the cell membrane; (iv) peroxidation decreases membrane fluidity; (v) cell shrinkage is induced by the reduction in membrane fluidity; (vi) owing to the cell shrinkage, abnormality in adhesion and reduction in total traction force are induced; (vii) owing to the reduction in membrane fluidity and abnormality in adhesion, fundamentally induced by intracellular ROS, the migratory activity of MNPs@SiO2(RITC)-treated hBM-MSCs decreases. The effect of this treatment on other cell functions will be addressed in future investigations.

Reviewer 2 Report
The paper Silica-coated magnetic nanoparticles decrease migratory activity of human bone marrow-derived mesenchymal stem cells via reduced membrane fluidity and impaired focal adhesion prepared by Tae Hwan Shin et al., present e very interesting study that deserve to be published after some minor improvement.
I strongly recommend to provide at least 3 characterization technique that prove the quality of nanoparticles.
After proper revision the paper deserve to be published.
Author Response
Answers to the comments
Reviewer # 2
The paper Silica-coated magnetic nanoparticles decrease migratory activity of human bone marrow-derived mesenchymal stem cells via reduced membrane fluidity and impaired focal adhesion prepared by Tae Hwan Shin et al., present e very interesting study that deserve to be published after some minor improvement.
I strongly recommend to provide at least 3 characterization technique that prove the quality of nanoparticles.
After proper revision the paper deserve to be published.
à We added the characterization techniques that prove the quality of nanoparticles in ‘Materials and Methods’ section as
2.1. MNPs@SiO2(RITC) and silica nanoparticles (NPs)
MNPs@SiO2(RITC) particles, composed of a ~9 nm cobalt ferrite core (CoFe2O3) chemically bonded to rhodamine isothiocyanate dye (RITC) and coated by a silica shell [4], were purchased from BITERIALS (Seoul, South Korea). Previously, these nanoparticles have been characterized for confirming their quality [45]. Size distribution and morphology are important factors determining the uniformity of nanoparticles and were analysed using electron and atomic microscopy [45]. Hydrodynamic size, polydispersity, and surface charge were determined using dynamic light scattering [46]. The purity and contents of nanoparticles are usually analysed using an X-ray based technique [45]. In this study, X-ray diffraction (XRD) analysis using a high power X-Ray diffractometer (Ultima III, Rigaku, Japan) confirmed the structure of MNPs@SiO2(RITC) (data not shown). The silica NPs were composed of identical materials and were of a similar size as the MNPs@SiO2(RITC) shell, and their biological effects were similar to those of MNPs@SiO2(RITC) [23,24,47,48]. The diameters of the MNPs@SiO2(RITC) and silica NPs were 50 nm, and the zeta potential of MNPs@SiO2(RITC) was between -40 to -30 mV [4,47]. A previous study determined ~ 105 particles of MNPs@SiO2(RITC) per cell in MNPs@SiO2(RITC)-treated MCF-7 cells using inductively coupled plasma atomic emission spectrometry [4]. Furthermore, in previous reports, the dosage was determined by measuring the fluorescence intensity of HEK293 cells treated with MNPs@SiO2(RITC) at concentrations ranging from 0.01 to 2.0 μg/μL for 12 h. The optimal concentration of MNPs@SiO2(RITC) was 0.1 µg/µL for in vitro use, whereas 1.0 µg/µL was the plateau concentration for cellular uptake [24]. Furthermore, MNPs@SiO2(RITC) concentrations ranging from 0 to 1.0 μg/μL have been used for MRI contrasting without toxicological effects on human cord blood-derived MSCs [49], and caused changes in gene expression and metabolic profiles similar to those of the control HEK293 cells at 0.1 µg/µL [24]. In addition, the uptake efficiency of MNPs@SiO2(RITC) almost plateaued at 1.0 µg/µL in HEK293 cells [24,25]. The dose-dependent fluorescence intensity of MNPs@SiO2(RITC)-labelled hBM-MSCs was similar to those of labelled HEK293 cells. In addition, the viability of human cord blood-derived MSCs was determined to assess the cytotoxic effect of MNPs@SiO2(RITC) after 24, 48, and 72 h of treatment with 0–1.0 µg/µL MNPs@SiO2(RITC); compared to the control group, no significant cytotoxic effect was observed [49]. Therefore, in this study, hBM-MSCs were treated with 0.1 µg/µL (low dose) MNPs@SiO2(RITC)or 1.0 µg/µL (high dose), similarly to previous reports [23,24,48].
Additional references
Mourdikoudis, S.; Pallares, R.M.; Thanh, N.T.K. Characterization techniques for nanoparticles: comparison and complementarity upon studying nanoparticle properties. Nanoscale 2018, 10, 12871-12934, doi:10.1039/c8nr02278j.
Fornaguera, C.; Lazaro, M.A.; Brugada-Vila, P.; Porcar, I.; Morera, I.; Guerra-Rebollo, M.; Garrido, C.; Rubio, N.; Blanco, J.; Cascante, A., et al. Application of an assay Cascade methodology for a deep preclinical characterization of polymeric nanoparticles as a treatment for gliomas. Drug Deliv 2018, 25, 472-483, doi:10.1080/10717544.2018.1436099.

Round 2
Reviewer 2 Report
all is ok now.